# MIXTURE-OF-CHANNELS: EXPLOITING SPARSE FFNS FOR EFFICIENT LLMS PRE-TRAINING AND INFERENCE

## ABSTRACT

Large language models (LLMs) have demonstrated remarkable success across diverse artificial intelligence tasks, driven by scaling laws that correlate model size and training data with performance improvements. However, this scaling paradigm incurs substantial memory overhead, creating significant challenges for both training and inference. While existing research has primarily addressed parameter and optimizer state memory reduction, activation memory—particularly from feed-forward networks (FFNs)—has become the critical bottleneck, especially when FlashAttention is implemented. In this work, we conduct a detailed memory profiling of LLMs and identify FFN activations as the predominant source to activation memory overhead. Motivated by this, we introduce **Mixture-of-Channels** (**MoC**), a novel FFN architecture that selectively activates only the Top-$K$ most relevant channels per token determined by SwiGLU's native gating mechanism. MoC substantially reduces activation memory during pre-training and improves inference efficiency by reducing memory access through partial weight loading into GPU SRAM. Extensive experiments validate that MoC delivers significant memory savings and throughput gains while maintaining competitive model performance.

## 1 INTRODUCTION

The rise of large language models (LLMs) has marked a paradigm shift in artificial intelligence, achieving unprecedented success across natural language processing, computer vision, decision making, and coding. The key drivers behind LLMs are scaling laws (Kaplan et al., 2020; Rae et al., 2021; Hoffmann et al., 2022), which demonstrate that model performance steadily improves with increased model size and expanded training data. However, scaling up models entails substantial memory costs, which increase training expenses, limit deployment on devices with restricted resources, and impede exploration of even larger and more capable models. Consequently, developing memory-efficient architectures and pre-training algorithms has become crucial for continued advancement in LLMs.

### 1.1 MOTIVATION

There has been extensive research on memory-efficient pre-training strategies. One prominent line of research focuses on parameter-efficient methods, which reduces the number of trainable parameters by leveraging low-rank approximations for model weights (Hu et al., 2022; Han et al., 2024; Lialin et al., 2023; Kamalakara et al., 2022; Miles et al.). An alternative approach focuses on compressing optimizer states while maintaining the number of trainable parameters. GaLore (Zhao et al., 2024) and its variants (He et al., 2024; Chen et al., 2024; Zhu et al., 2024a) achieve this by leveraging low-rank gradients to compute first- and second-order moments. Adafactor (Shazeer & Stern, 2018), Adam-mini (Zhang et al., 2024a), and Apollo (Zhu et al., 2024a) reduce redundancy in the state variables of the Adam optimizer. Although improving pre-training efficiency, these techniques do not address the memory overhead in activation storage. Experimental evidence (Touvron et al., 2023; Grattafiori et al., 2024) suggests that, especially during pre-training with large batch sizes and long sequences, activation memory constitutes a substantial, often dominant, portion of the total memory footprint. This highlights the paramount importance of developing activation-efficient techniques.

In Transformer-based LLMs, activation memory primarily originates from the attention mechanism and feed-forward networks (FFNs). Significant advances such as FlashAttention (Dao et al., 2022;

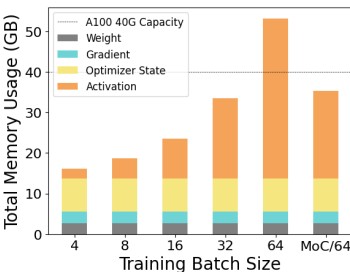

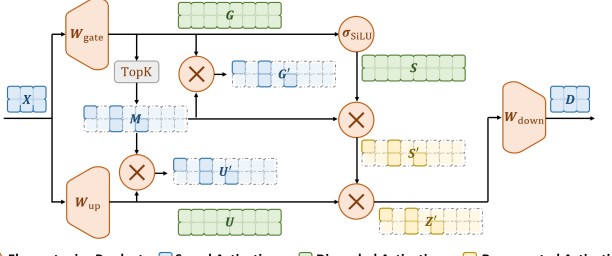

Figure 1: Memory breakdown of pre-training LLaMA-2 with a fixed sequence length of 256 and various batch size choices.

Figure 2: An illustration of the Mixture-of-channels (MoC) architecture and its modification to the standard SwiGLU FFN. Components painted in blue are stored as activations during the forward pass, and those in yellow will be efficiently recomputed during backward pass.

Dao, 2023; Shah et al., 2024) have mitigated the quadratic memory complexity $\mathcal{O}(s^2)$ of standard attention operations with respect to sequence length $s$, thereby reducing it to linear complexity $\mathcal{O}(s)$. This breakthrough substantially alleviates the memory overhead of attention mechanisms in long-context settings. Consequently, FFN activation memory has emerged as the dominant bottleneck, see our detailed profiling in Figure 1. However, the challenge of efficiently compressing FFN's activation memory without incurring severe performance degradation remains largely underexplored.

## 1.2 MAIN RESULTS AND CONTRIBUTIONS

In mainstream LLM architectures, FFNs typically employ GeLU (Hendrycks & Gimpel, 2016) or SwiGLU (Shazeer, 2020) as the activation function. In this work, we aim to develop activation-efficient FFN architectures tailored to such widely used activation functions, with the goal of reducing activation memory with negligible sacrifice of performance. Our contributions are as follows.

**C1. Detailed activation profiling.** We provide a detailed memory profile of LLM activations with FlashAttention applied. Both theoretical analysis and empirical evidence confirm that FFN activations have become the dominant activation memory bottleneck. This finding strongly motivates the development of activation-efficient FFN architectures.

**C2. Key activation patterns.** We analyze the value distributions of activations generated by SwiGLU functions in FFNs. Our findings show that more than 70% of activation values are near zero or slightly negative. This indicates that, for a given input token, only a small subset of activation channels are highly active and contribute meaningfully to the backward computation.

**C3. Activation-efficient FFN architecture.** We propose **Mixture-of-Channels** (**MoC**)—a novel FFN architecture that selectively activates only a mixture of the top-$K$ task-relevant channels for each token. MoC leverages SwiGLU's inherent gating signal to rank channel importance. During pre-training, MoC reduces activation memory by storing only the truly-active channels; during inference, it lowers memory access overhead by loading only the required weights into GPU SRAM at each decoding step, resulting in a substantial throughput improvement.

**C4. System-aware implementations.** We develop a set of hardware-aware kernels to further accelerate both pre-training and inference. First, we implement a batched Top-$K$ filtering operator using the RAFT kernel library (Rapidsai, 2022), achieving significant speedup over PyTorch's native implementation. Second, we design custom forward and backward kernels for the MoC architecture using Triton, delivering comparable training throughput and approximately a $1.4\times$ decoding speedup in the FFN compared to standard LLaMA-style Transformer implementations.

We conduct extensive experiments to validate the benefits of MoC. In pre-training, MoC demonstrates substantial memory savings while outperforming existing memory-efficient methods in performance. During inference, MoC achieves a 1.13× speedup in end-to-end decoding latency.

## 1.3 COMPARISON WITH EXISTING SPARSE-ACTIVATION APPROACHES

**Sparse pre-training.** Recent studies have identified a distinctive property of pre-trained Transformers: their intermediate layers exhibit sparse activation patterns (Zhang et al., 2022; Dong et al., 2023;

Mirzadeh et al., 2023). While this phenomenon has been harnessed in several post-training methods to accelerate inference (Liu et al., 2023; Song et al., 2024; Alizadeh et al., 2024), its potential for reducing activation memory during the pre-training phase remains relatively underexplored. Earlier work investigated sparse training in CNN-based architectures such as ResNet and VGG (Raihan & Aamodt, 2020; Jiang et al., 2022); however, these studies do not address sparsity patterns in LLMs. Reference Zhang et al. (2024b) proposed switchable sparse-dense learning for LLM pre-training. However, this method fails to reduce peak activation memory due to its reliance on periodic dense learning phases. In contrast, our proposed MoC framework maintains persistent sparsity throughout training, thereby significantly lowering peak memory consumption. A recent effort Wang et al. (2024) replaced SwiGLU with SquareReLU to induce greater activation sparsity during LLM pre-training. While this modification enhances sparsity, our approach retains SwiGLU—a widely adopted activation function in LLMs—ensuring compatibility with established architectural practices.

**Sparse inference.** Recent advances in efficient inference for LLMs have extensively explored sparse activation patterns, primarily through two approaches: attention-based methods that dynamically manage Key-Value memory retention by selectively preserving critical pairs (Xiao et al., 2023; Tang et al., 2024; Zhang et al., 2023b), and FFN-based methods that target sparsity in feed-forward networks (our work falls into this line). However, existing FFN-based approaches operate on originally dense LLMs, requiring post-training modifications (Mirzadeh et al., 2023; Wang et al., 2024) like activation function redesign or targeted pruning to induce sparsity, while often depending on dynamic thresholding mechanisms (Lee et al., 2024; Liu et al., 2024a) whose huristic nature introduces performance variability. In contrast, our MoC FFN addresses these limitations through its intrinsic Top-$K$ sparsity pre-training mechanism, which guarantees sparsity patterns without additional post-hoc modifications. This architectural innovation eliminates the heuristic threshold determination while simultaneously preserving both inference acceleration and model integrity.

## 1.4 ORTHOGONALITY TO EXISTING ACTIVATION-EFFICIENT APPROACHES

**FlashAttention.** FlashAttention (Dao et al., 2022; Dao, 2023; Shah et al., 2024) is an optimized attention implementation that strategically reorganizes computations to minimize memory access overhead and maximize GPU memory bandwidth utilization. Through its tiled computation and fused kernel design, it achieves significant speed improvements and memory reduction specifically within Transformer's attention module. Targeting the memory efficiency of the FFN component, our MoC architecture is fundamentally orthogonal to FlashAttention. In our experiments, we combine these two techniques to deliver compounded activation efficiency gains across the entire LLM architecture.

**Mixed-precision methods.** Mixed-precision training has become indispensable for efficiently scaling modern LLMs, reducing memory usage and accelerating computation. The seminal work by Micikevicius et al. (2017) demonstrated that storing activations in FP16 enables stable training with reduced activation memory overhead. Recent studies such as GPTQ (Frantar et al., 2022) and AWQ (Lin et al., 2024) have quantized activations to FP8/int4 formats in LLMs post-training with less than 1% performance degradation. To pre-train LLMs, quantization-aware training (QAT) methods such as BitNet (Wang et al., 2023) and Quest (Panferov et al., 2025) enable 4-bit quantization for activations. Mixed-precision can also be applied to our MoC structure.

**Gradient checkpointing.** Several system-level techniques have been proposed to improve activation memory efficiency. Gradient checkpointing (Chen et al., 2016) reduces memory consumption by selectively recomputing activations during backpropagation instead of storing them throughout the forward pass, at the cost of increased computation. Other approaches Ren et al. (2021); Zhang et al. (2023a) reduce GPU memory usage by offloading data to non-GPU resources, introducing additional communication overhead. As a FFN architecture, MoC is compatible with gradient chekpointing.

Additional related work on parameter- and optimizer-efficient methods is discussed in Appendix A.

## 2 ACTIVATION MEMORY PROFILING

This section provides a detailed analysis of activation memory during the pre-training stage of LLMs represented by LLaMA-2 (Touvron et al., 2023) with FlashAttention applied.

**LLM structure.** Each Transformer layer consists of two main modules: Multi-Head Self-Attention (MHSA) and a Feed-Forward Network (FFN). Pre-normalization using RMSNorm is applied before both the MHSA and FFN modules to enhance training stability. Residual connections are added after both MHSA and FFN components to facilitate the training of deep networks. The FFN module is implemented as a two-layer MLP with SwiGLU activation functions. See illustrations in Figure 3.

**Theoretical analysis.** We denote the training batch size, sequence length, and model hidden dimension by $b$, $s$, and $d$, respectively. Let $h$ represent the number of attention heads, define $d_h = d/h$ as the hidden dimension per head, and let $d_{\text{ffn}}$ be the FFN hidden dimension. Below, we present a detailed theoretical analysis of activation memory during the pre-training phase of LLMs.

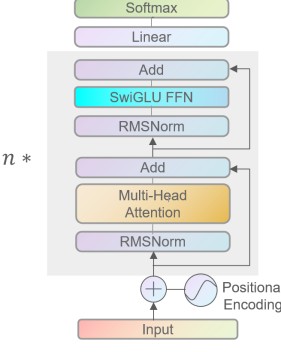

$n *$

Figure 3: LLaMA-2 architecture.

- **MHSA.** For each input $X \in \mathbb{R}^{s \times d}$ to the MHSA module, we first need to compute and store, for each attention head $i$, the following activations:

$$Q_i = XW_Q^i \in \mathbb{R}^{s \times d_h}, \quad K_i = XW_K^i \in \mathbb{R}^{s \times d_h}, \quad V_i = XW_V^i \in \mathbb{R}^{s \times d_h},$$

where $W_Q^i, W_K^i, W_V^i \in \mathbb{R}^{d \times d_h}$ are weights for the $i$-th attention head. We next store

$$A_i = \text{FlashAttention}(Q_i, K_i, V_i) \in \mathbb{R}^{s \times d_h} \quad \text{and} \quad O = AW_o \in \mathbb{R}^{s \times d},$$

where $A = [A_1; \cdots ; A_h] \in \mathbb{R}^{s \times d}$ is the concatenated attention output and $W_o \in \mathbb{R}^{d \times d}$ is the weight. Since FlashAttention leverages kernel fusion and recomputation techniques, it eliminates the need to store intermediate variables such as attention weights or softmax outputs during the forward pass. As a result, MHSA needs to store $Q, K, V, A$ and $O$, amounting to a total of $5sd$ activations. For a batch size of $b$, this corresponds to an activation memory footprint of $5bsd$.

- **FFN.** For each input $X \in \mathbb{R}^{s \times d}$ to the FFN module, we first compute and store:

$$G = XW_{\text{gate}} \in \mathbb{R}^{s \times d_{\text{ffn}}}, \quad U = XW_{\text{up}} \in \mathbb{R}^{s \times d_{\text{ffn}}}, \tag{1}$$

where $W_{\text{gate}}, W_{\text{up}} \in \mathbb{R}^{d \times d_{\text{ffn}}}$ are the weights corresponding to the gating and up-sampling branches in the SwiGLU activation. We then compute and store

$$S = \text{SiLU}(G) \in \mathbb{R}^{s \times d_{\text{ffn}}}, \quad Z = S \odot U \in \mathbb{R}^{s \times d_{\text{ffn}}}, \quad D = ZW_{\text{down}} \in \mathbb{R}^{s \times d}, \tag{2}$$

where $W_{\text{down}} \in \mathbb{R}^{d_{\text{ffn}} \times d}$ is the down-sampling weight. As a result, the FFN module needs to store $U, G, S, Z$ and $D$, amounting to a total of $(4d_{\text{ffn}} + d)s$ activations. For a batch size of $b$, this corresponds to an activation memory footprint of $b(4d_{\text{ffn}} + d)s$. Since the typical choice of $d_{\text{ffn}}$ is $\frac{8d}{3}$, the activation memory is around $11.67bsd$.

- **RMSNorm.** For each row $x \in \mathbb{R}^d$ of the input $X \in \mathbb{R}^{s \times d}$, we compute and store $y = \frac{x}{\text{RMS}(x)} \odot g \in \mathbb{R}^d$ where $\text{RMS}(x) = (\frac{1}{d} \sum_{i=1}^d x_i^2 + \epsilon)^{1/2}$ and $g \in \mathbb{R}^d$ is a learned weight vector. Since there are $s$ rows in total, this results in $sd$ activations per RMSNorm layer. As there are two RMSNorm layers—one before the MHSA module and one before the FFN module—the total number of activations is $2sd$. For a batch size of $b$, the activation memory is $2bsd$.

- **Residual connections.** Each residual connection adds the module output to its input. Given input $X \in \mathbb{R}^{s \times d}$, we compute $Z = X + Y \in \mathbb{R}^{s \times d}$, where $Y$ is the module output. To support backpropagation, the input $X$ must be stored, requiring $sd$ activations. Since each transformer layer has two residual connections, the total is $2bsd$ activations.

Following the above analysis, we finally decompose the activation memory into

*Layer Activation = Attention $5bsd$ + FFN $11.67bsd$ + RMSNorm $2bsd$ + Residual $2bsd$*

This shows that the FFN dominates activation memory when FlashAttention is applied. Moreover, FFNs primarily consist of large matrix multiplications, which are difficult to optimize without compromising model performance.

|  |  | LLaMA (350M) | LLaMA (1.3B) |
|---|---|---|---|
| Per-Layer (×24) | **Attention** | 177M | 336M |
|  | **FFN** | 400M | 791M |
|  | **Others** | 68M | 134M |
|  | **LLM head** | 2.16G | 2.16G |
|  | **Total** | 17.64G | 32.4G |

**Empirical profiling.** The right table presents profiling when batchsize is 64 and sequence length is 256. In both models, the FFN-to-Attention memory ratio is approximately 2.3, which closely matches our theoretical analysis.

## 3 THE MIXTURE-OF-CHANNELS ARCHITECTURE

We introduce **Mixture-of-Channels (MoC)**, an activation-efficient FFN architecture designed to substantially reduce memory usage during pre-training and accelerate decoding during inference.

### 3.1 SiLU ACTIVATION PATTERN.

SiLU (also known as $\text{Swish}_1$) is an activation function commonly used in the FFN layers of LLMs, see (1) and (2). SiLU is defined as

$$\text{SiLU}(x) = \frac{x}{1 + \exp(-x)}, \tag{3}$$

which is illustrated in Figure 4. Notably, when $x \geq 0$, $\text{SiLU}(x)$ approaches $x$, indicating strong activation; conversely, for negative $x$, the output tends toward zero, implying that the input is largely suppressed. This nonlinear behavior naturally induces a degree of sparsity in the FFN outputs. Motivated by this observation, we investigate how many channels remain active after applying SiLU in pre-trained LLMs.

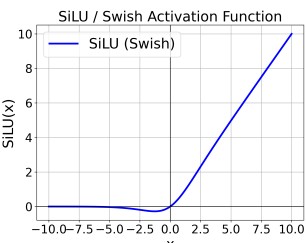

Figure 4: SiLU activation.

Figure 5 presents histograms of pre-SiLU and post-SiLU activations from various layers of the pre-trained LLaMA-2 model. As shown in Figures 5a–5c, approximately 70% of the inputs to the SiLU function are negative. Based on the SiLU expression illustrated in Figure 4, this results in about 70% of the SiLU outputs being close to zero (see Figures 5d–5f). This implies that a significant portion of the channels are effectively suppressed after SiLU activation, with only around 30% remaining active. This key observation motivates the design of activation-efficient FFN architectures. Similar observations also hold for MoE models, see the results in Appendix C.

### 3.2 MIXTURE-OF-CHANNELS ARCHITECTURE

Recalling the FFN structure listed in (1)–(2), for input $X \in \mathbb{R}^{s \times d}$, we first compute

$$G = XW_{\text{gate}}, \qquad U = XW_{\text{up}}. \tag{4}$$

According to the SiLU activation pattern in Section 3.1, most elements in $G$ are negative and are suppressed by the SiLU activation function. To reduce activation memory, we propose retaining only the large positive elements of $G$ and masking out the rest during pre-training. To this end, we introduce a binary mask matrix $M = \text{TopK}(G) \in \mathbb{R}^{s \times d_{\text{ffn}}}$, in which the entries of $M$ satisfy:

$$M_{ij} = \begin{cases} 1, & \text{if } G_{ij} \text{ is among the top-}K \text{ largest values in row } i \text{ of } G, \\ 0, & \text{otherwise.} \end{cases} \tag{5}$$

Here, the top-$K$ selection is applied in a row-wise manner, so each row of $M$ contains exactly $K$ ones. Note that $M$ retains the top-$K$ largest values (not the largest in absolute value), focusing on strongly active channels in each row of $G$. We then apply the mask matrix $M$ to sparsify the intermediate representations $S$ and $Z$ in equation (2) as follows:

$$S = \text{SiLU}(G), \quad S' = S \odot M, \quad Z' = S' \odot U, \quad D = Z'W_{\text{down}}. \tag{6}$$

This completes the formulation of the proposed architecture. Since it selectively utilizes a mixture (not all) of the channels from the original $S$ and $Z$ matrices, we refer to it as the Mixture-of-Channels (MoC) architecture. Figure 2 illustrates the MoC architecture and its forward process.

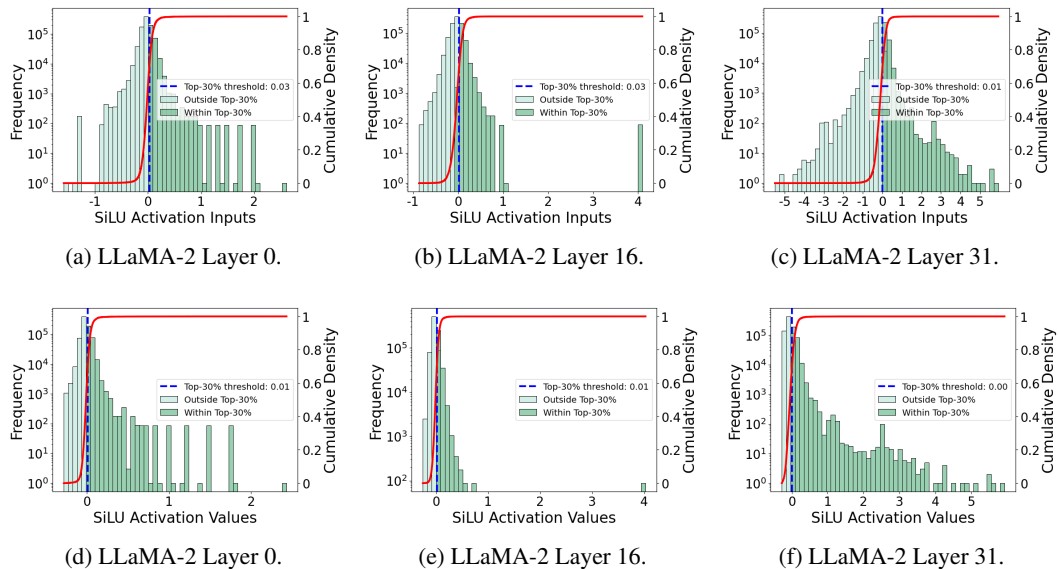

(a) LLaMA-2 Layer 0.    (b) LLaMA-2 Layer 16.    (c) LLaMA-2 Layer 31.

(d) LLaMA-2 Layer 0.    (e) LLaMA-2 Layer 16.    (f) LLaMA-2 Layer 31.

Figure 5: Histograms of pre-SiLU and post-SiLU activations from different layers of LLaMA-2. Subfigures (a), (b), and (c) correspond to the pre-SiLU activations, while subfigures (d), (e), and (f) show the post-SiLU activations. The blue dashed line marks the threshold for the top 30% of activations by value, and the red curve represents the cumulative distribution.

## 4    ACTIVATION-EFFICIENT PRE-TRAINING WITH MIXTURE-OF-CHANNELS

**Backward propagation.** We now derive the backward propagation step for the MoC architecture. Let $\nabla_D$ denote the gradient of the loss with respect to the output $D$. The backward propagation is:

$$\nabla_{W_{\text{down}}} = (Z')^\top \nabla_D, \qquad \nabla_{Z'} = \nabla_D W_{\text{down}}^\top, \tag{7a}$$

$$\nabla_{S'} = (U \odot M) \odot \nabla_{Z'}, \qquad \nabla_U = S' \odot \nabla_{Z'}, \tag{7b}$$

$$\nabla_S = \nabla_{S'}, \qquad \nabla_G = \nabla_{S'} \odot (\nabla \text{SiLU})(G), \tag{7c}$$

$$\nabla_{W_{\text{gate}}} = X^\top \nabla_G, \qquad \nabla_{W_{\text{up}}} = X^\top \nabla_U, \tag{7d}$$

$$\nabla_X = \nabla_G W_{\text{gate}}^\top + \nabla_U W_{\text{up}}^\top, \tag{7e}$$

where $\nabla \text{SiLU}(\cdot)$ in (7c) is the gradient operator of the SiLU activation function defined in (3). To facilitate the backward propagation steps described above, we need to store the activations $Z' = S' \odot U = S \odot M \odot U = Z \odot M$ from (7a), as well as $U \odot M$ and $S' = S \odot M$ from (7b). Moreover, since $\nabla_{S'} = \nabla_S = U \odot M \odot \nabla_{Z'}$ exhibits a sparse pattern, it suffices to store only $G \odot M$ in (7c) instead of the full matrix $G$. Table 1 compares the activation memory of the standard FFN and the MoC architecture, where each row of $M$ retains the top 20% of its elements, as used in our experiments. When set $d_{\text{ffn}} = \frac{8d}{3}$, MoC substantially reduces activation memory usage from $11.67bsd$ to $3.67bsd$, making the FFN activation even smaller than that of FlashAttention.

**MoC + GCP.** Recall from (2) that $S = \text{SiLU}(G)$ and $Z = S \odot U$, both of which involve inexpensive element-wise operations. As a result, it is unnecessary to store $S$ and $Z$ in memory; instead, they can be recomputed from $G$ during backpropagation. This technique is known as gradient checkpointing (GCP). Table 1 compares the activation memory and additional computation overhead introduced by gradient checkpointing (GCP) for the standard FFN and the MoC architecture. MoC+GCP achieves substantial memory savings with significantly lower computational overhead compared to FFN+GCP.

**MoC's Expressive Power.** To quantify the expressive capacity of MoC despite its efficiency-oriented design, we analyze its relationship to the standard FFN introduced in Section 2. Let $\mathcal{H}_{d_{\text{ffn}}}$ denote the class of operators $\mathbb{R}^d \to \mathbb{R}^d$ implemented by a standard FFN with hidden dimension $d_{\text{ffn}}$, and let $\mathcal{H}_{d_{\text{moc}}}^{a:b}$ denote the class of operators $\mathbb{R}^d \to \mathbb{R}^d$ realized by an MoC$_{a:b}$ model (where $b \mid d_{\text{moc}}$ and $a \leq b$) with hidden dimension $d_{\text{moc}}$. Here, $a{:}b$ denotes the configuration of MoC's Grouped Top-$K$

Table 1: Activation memory comparison during pre-training. "GCP Comp" refers to computational overhead introduced by gradient checkpointing. Memory values in parentheses are computed using $d_{\text{ffn}} = \frac{8d}{3}$.

|  | MoC | MoC+GCP | FFN | FFN+GCP |
|---|---|---|---|---|
| **Stored Activations** | $G \odot M$ 
 $U \odot M$ 
 $S \odot M$ 
 $Z \odot M$ 
 $M$ and $D$ | $G \odot M$ 
 $U \odot M$ 
 $-$ 
 $-$ 
 $M$ and $D$ | $G$ 
 $U$ 
 $S$ 
 $Z$ 
 $D$ | $G$ 
 $U$ 
 $-$ 
 $-$ 
 $D$ |
| **Memory Cost** | $bsd_{\text{ffn}} + bsd$ 
 $(3.67bsd)$ | $0.6bsd_{\text{ffn}} + bsd$ 
 $(2.6bsd)$ | $4bsd_{\text{ffn}} + bsd$ 
 $(11.67bsd)$ | $2bsd_{\text{ffn}} + bsd$ 
 $(6.33bsd)$ |
| **GCP Comp.** | $-$ | $0.4bsd_{\text{ffn}}$ 
 $(1.07bsd)$ | $-$ | $2bsd_{\text{ffn}}$ 
 $(5.33bsd)$ |

selection strategy, in which $a$ elements are selected from each contiguous group of $b$ entries. We establish the following theorem.

**Theorem 1.** *For all $a \leq b$ and $d_{\text{ffn}} \in \mathbb{N}^*$, it holds that (Proof is in Appendix B)*

$$\mathcal{H}_{d_{\text{ffn}}} \subseteq \mathcal{H}_{d_{\text{moc}}}^{a:b}, \quad where \quad d_{\text{moc}} = b\lceil d_{\text{ffn}}/a \rceil.$$

**Remark.** Theorem 1 provides a lower bound on MoC's expressive capacity. In particular, when $b/a = \Theta(1)$, any standard FFN can be emulated by an MoC$_{a:b}$ model with only a constant-factor increase in parameters. Hence, MoC$_{a:b}$ matches the expressive power of a standard FFN up to constant factors in model size.

**Hardware-aware implementation.** While the MoC architecture theoretically provides computation savings, realizing this benefit in practice requires careful system-level optimization. In particular, unstructured sparsity does not translate to actual computational gains on modern accelerators (e.g., GPUs and TPUs), as they are not designed to exploit irregular sparsity patterns efficiently. Additionally, the top-$K$ selection introduces computational overhead, potentially offsetting the theoretical advantages. We made two efforts to address these issues:

|  | **Standard FFN (ms)** | **MoC using optimized kernels (ms)** |
|---|---|---|
| **Forward** | 20.2 | 21.1 |
| **Backward** | 21.6 | 22.8 |
| **Total** | 41.8 | 43.9 |

- **Customized MoC kernels.** We develop a hardware-aware implementation by customizing a batched top-$K$ operator using the RAFT library (Rapidsai, 2022) and designing fused Triton kernels to accelerate intermediate computations. These optimizations enable MoC to achieve computation efficiency comparable to that of standard FFNs, see the above tables.

- **MoC with structured 2:8 sparsity.** We implement structured 2:8 sparsity—where only 2 out of every 8 consecutive weights are retained—a format natively supported by NVIDIA Ampere and Hopper architectures. This structured sparsity unlocks the computational efficiency of MoC on compatible hardware, bridging the gap between theoretical and practical performance. We denote MoC$_{2:8}$ as the MoC architecture supports structured 2:8 sparsity.

## 5 ACCELERATED INFERENCE WITH MIXTURE-OF-CHANNELS

Now we consider the inference process with MoC architecture. Given an input token $x \in \mathbb{R}^{1 \times d}$, the MoC inference will proceed as follows.

- (Linear projection) Compute $g = xW_{\text{gate}} \in \mathbb{R}^{1 \times d_{\text{ffn}}}$ and $u = xW_{\text{up}} \in \mathbb{R}^{1 \times d_{\text{ffn}}}$.

- (Top-$K$ masking) Construct a binary mask $m \in \{0, 1\}^{1 \times d_{\text{ffn}}}$, where:

$$m_j = \begin{cases} 1, & \text{if } g_j \text{ is among the top-}K \text{ values in } g, \\ 0, & \text{otherwise.} \end{cases}$$

- (Nonlinearity and sparsification) Compute $s = \text{SiLU}(g), s' = s \odot m, z' = s' \odot u$.

Table 2: Comparison of different memory-efficient algorithms during the pre-training of various LLaMA-based Transformer models on the C4 dataset. We report validation perplexity alongside total memory usage, which includes model weights, gradients, optimizer states, and activations. Perplexity results for GaLore and SLTrain are taken from (Zhao et al., 2024; Han et al., 2024). Constant $K$ is the number of activated channels. We typically set $K = 0.5d_{model}$, which is 18.75% of of the channel dimension $d_{ffn} = 8d_{model}/3$.

|  | **60M** | **130M** | **350M** | **1B** |
|---|---|---|---|---|
| FFN-based LLM + AdamW | 30.44 (38.3G) | 23.92 (54.1G) | 18.26 (52.5G) | 15.56 (56.6G) |
| GaLore | 34.88 (38.1G) | 25.36 (53.9G) | 18.95 (51.7G) | 15.64 (52.5G) |
| SLTrain | 34.15 (33.6G) | 26.04 (59.4G) | 19.42 (69.1G) | 16.14 (70.0G) |
| MoC | **30.59** (21.8G) | **24.02** (41.7G) | **18.57** (34.6G) | 15.80 (41.9G) |
| MoC$_{2:8}$ | 31.02 (22.1G) | 24.12 (42.3G) | 18.68(36.4G) | **15.62** (42.7G) |
| batch size per GPU | 256 | 256 | 128 | 64 |
| $K/d_{model}$ | 128 / 256 | 384 / 768 | 512 / 1024 | 1024 / 2048 |
| Training Tokens | 1.1B | 2.2B | 6.4B | 13.1B |

Table 3: Zero-shot evaluation results on downstream tasks.

|  | **MMLU** | **ARC-Easy** | **ARC-Challenge** | **PIQA** | **TruthfulQA-MC2** | **Avg.** |
|---|---|---|---|---|---|---|
| LLaMA-1B | **0.231** | 0.426 | 0.237 | **0.678** | 0.398 | 0.394 |
| MoC-1B | 0.230 | 0.416 | 0.237 | 0.636 | **0.455** | 0.395 |
| MoC$_{2:8}$-1B | 0.230 | **0.432** | **0.242** | 0.655 | 0.454 | **0.403** |

- (Final output) Compute the output $d = z'W_{down} = \sum_{j \in \mathcal{C}} z'_j w_j \in \mathbb{R}^{1 \times d}$ where $w_j$ denotes the $j$-th row of $W_{down}$, and $\mathcal{C}$ is the set of active (i.e., unmasked) channels.

It is observed that the inference relies only on the subset of active channels selected per token; this selection is driven by SwiGLU's native gating mechanism, which identifies the top-$K$ most relevant channels for each input token.

**Accelerated inference via sparse activation.** The decoding latency of LLMs is primarily IO-bounded by loading weights from GPU HBM to SRAM, and our sparse MoC architecture delivers substantial inference-time speedups by alleviating unnecessary memory access costs (MACs). In a standard FFN with hidden dimension $d_{ffn}$, each token incurs two dense projections ($xW_{gate}$ and $xW_{up}$) plus one down-projection ($zW_{down}$), for a total of $2d \times d_{ffn} + d_{ffn} \times d = 3\,d\,d_{ffn}$ MACs per token. By contrast, MoC retains only the top-$K$ channels in each of $u$ and $z'$, so we need only $K$ columns of $W_{up}$ and $K$ rows of $W_{down}$, yielding $d\,d_{ffn} + 2\,K\,d$ MACs per token. When $K \ll d_{ffn}$, this represents an $\mathcal{O}(K/d_{ffn})$ reduction in MACs, contributing to much faster inference.

**Intrinsic sparse pattern.** One notable advantage of the MoC architecture is that it naturally introduces an intrinsic sparsity pattern during pre-training, as only the top-$K$ channels are activated for each input. In contrast, existing sparse inference methods built on dense LLMs typically require post-hoc pruning or distillation to achieve comparable efficiency (Xiao et al., 2023; Tang et al., 2024; Zhang et al., 2023b; Mirzadeh et al., 2023; Wang et al., 2024; Lee et al., 2024; Liu et al., 2024a), which can degrade performance or necessitate additional training steps.

# 6 EXPERIMENTS

We evaluate the pre-training performance and inference efficiency of LLMs using the MoC architecture. All experiments are conducted on NVIDIA A800 GPUs with 80 GB of VRAM.

## 6.1 MEMORY REDUCTION AND TRAINING PERFORMANCE

**Pre-training LLaMA on C4.** To evaluate the expressive power of models using the MoC architecture, we pre-train LLaMA-based large language models with various memory-efficient strategies on the C4 dataset (Raffel et al., 2020). We closely follow the experimental setup in GaLore (Zhao et al., 2024) to ensure fair comparisons with several baselines, including vanilla AdamW applied to standard FFN-

Table 4: Inference latency($\mu$s) of a single FFN layer for a single batch size at each decoding step.

|  | Gate Proj. | Up&Down Proj. | Top-$K$ | Other | Total | Rel. Speedup |
|---|---|---|---|---|---|---|
| Standard FFN | 24 | 56.5 | **0** | 4.8 | 85.3 | 1.0$\times$ |
| MoC | 24 | **18.2** | 15 | **4.6** | 61.8 | 1.38$\times$ |
| MoC$_{2:8}$ | 24 | **18** | 9.4 | **4.6** | **56.0** | **1.52$\times$** |

Table 5: End-to-end inference speedup, measured by decoding throughput.

|  | Decode Latency | Decode Throughput | Rel. Speedup |
|---|---|---|---|
| Vanilla LLaMA | 4.75 ms | 210.45 tokens/sec | 1.0$\times$ |
| MoC | **4.20 ms** | **238.10 tokens/sec** | **1.13$\times$** |

based LLMs, GaLore, which aims to reduce optimizer memory usage, and SLTrain (Han et al., 2024), which focuses on reducing weight memory. Our evaluation considers both model performance and pre-training memory consumption. All models are trained using the AdamW (Loshchilov & Hutter, 2019) optimizer with a cosine annealing learning rate scheduler, with all other hyperparameters aligned to those of the baseline methods. FlashAttention is enabled, activation checkpointing is disabled, and the BF16 data format is employed. Additional experimental details are provided in Appendix C. (**Performance and Memory**) The training results are presented in Table 2. Both MoC and MoC$_{2:8}$ achieve the best performance among all baselines while significantly reducing memory consumption by substantially reducing activations—the dominant contributor to overall memory usage. (**Throughput**) We measure the throughput of different models by pre-training LLaMA-350M and LLaMA-1B using batch sizes of 128 and 64, respectively, on a single NVIDIA A800 GPU. The results in Appendix C demonstrates that MoC achieves training efficiency comparable to that of the vanilla LLaMA model due to our hardware-aware kernel implementation. (**Generalization**) As shown in Table 3, our model demonstrates strong performance across a broad range of tasks relative to the baseline LLaMA model. However, a modest performance gap persists in specific benchmarks such as PIQA (Bisk et al., 2020), suggesting limitations in commonsense reasoning capabilities.

### 6.2 INFERENCE ACCELERATION

**FFN inference speedup.** We measure the latency of a single FFN layer in LLaMA-1.3B for a batch size of one at each decoding step, with results reported in Table 4. To ensure a fair comparison with the vanilla FFN, both implementations are optimized using `torch.compile()`. As discussed earlier, MoC leverages the sparsity pattern in SiLU activations, which significantly accelerates both the up-projection and down-projection computations. This effectively offsets the overhead introduced by the top-$K$ selection. Table 4 shows that MoC achieves 1.38$\times$ speedup compared to FFN.

**End-to-end inference speedup**. We also measure the end-to-end decoding throughput of LLaMA 1.3B and MoC 1.3B, with results reported in Table 5. The input batch size is 1, and the prompt and generation length are 128. Both models are optimized using `torch.compile()`.

**MoC with 2:8 sparsity.** In Section 5, we propose MoC with 2:8 sparsity, which retains only the top-2 outputs among every 8 activations. As shown in Tables 2, 3, and 18, MoC$_{2:8}$ achieves training efficiency and performance comparable to denser variants and vanilla LLaMA. Furthermore, according to the decoding latency of the FFN reported in Table 4, MoC$_{2:8}$ offers superior inference performance by aligning with the coalesced memory access patterns of GPU hardware.

## 7 CONCLUSION AND LIMITATIONS

We propose Mixture-of-Channels (MoC), a novel feedforward network (FFN) architecture that activates only the Top-$K$ most relevant channels per token, guided by the native gating mechanism of SwiGLU. MoC significantly reduces activation memory during pre-training and improves inference efficiency by reducing memory access costs through selective channel activation. One limitation of MoC is that its inference acceleration relies on the sparse structure present under limited batch sizes, a challenge commonly encountered in activation sparsity-based inference acceleration methods.

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

## A    MORE RELATED WORKS

**Optimizer-efficient approaches.** An alternative strategy for reducing memory usage focuses on compressing optimizer states while retaining the full set of trainable parameters. GaLore (Zhao et al., 2024) achieves this by projecting the gradient matrix onto a low-dimensional subspace and using the compressed gradient to compute both first- and second-order moments. Although the projection matrix is typically obtained via Singular Value Decomposition (SVD) of the true gradient (Zhao et al., 2024), several more computationally efficient alternatives have been proposed, including random projections (He et al., 2024; Hao et al., 2024), norm-based scaling (Chen et al., 2024), and error feedback (Robert et al., 2024). Another line of research aims to reduce redundancy in the optimizer states of Adam. For example, Adam-mini (Zhang et al., 2024a) compresses the second-order momentum, while Apollo (Zhu et al., 2024a) eliminates all optimizer states by using approximate gradient scaling.

**Parameter-efficient approaches.** A promising direction for memory-efficient training involves parameter-efficient approaches, which reduce the number of trainable parameters and, consequently, the memory overhead associated with storing optimizer states. For example, LoRA (Hu et al., 2022) and its variants Liu et al. (2024b); Hayou et al. (2024); Malinovsky et al. (2024) constrain updates to a low-rank subspace of each weight matrix. While these methods significantly reduce memory consumption, the limited number of trainable parameters can sometimes lead to degraded model performance (Biderman et al., 2024). To address this issue, recent work proposes employing multiple LoRA modules to enable effectively high-rank updates (Lialin et al., 2023; Xia et al., 2024). However, in pre-training settings, this strategy still relies on an initial full-rank training phase as a warm-up before transitioning to low-rank updates (Lialin et al., 2023), thereby limiting its overall memory efficiency.

## B    PROOF OF THEOREM 1

In this section, we provide the detailed proof of Theorem 1.

*Proof of Theorem 1.* Given $a \leq b$ and $d_{\text{ffn}} \in \mathbb{N}^*$, it suffices to prove that for $\forall\ f \in \mathcal{H}_{d_{\text{ffn}}}$, we have $f \in \mathcal{H}_{d_{\text{moc}}}^{a:b}$, where $d_{\text{moc}} = b\lceil d_{\text{ffn}}/a \rceil$. According to the definition of $\mathcal{H}_{d_{\text{ffn}}}$, there exist weight matrices $W_{\text{gate}}, W_{\text{up}} \in \mathbb{R}^{d \times d_{\text{ffn}}}$ and $W_{\text{down}} \in \mathbb{R}^{d_{\text{ffn}} \times d}$ such that $f$ can be parameterized as $f = \text{FFN}(W_{\text{gate}}, W_{\text{up}}, W_{\text{down}})$. Let $k = \lceil d_{\text{ffn}}/a \rceil$ and $r = ka - d_{\text{ffn}}$, it holds that $0 \leq r < a$ and $d_{\text{moc}} = kb$. We construct matrices $W'_{\text{gate}}, W'_{\text{up}} \in \mathbb{R}^{d \times d_{\text{moc}}}$ and $W'_{\text{down}} \in \mathbb{R}^{d_{\text{moc}} \times d}$ as follows:

$$W'_{\text{gate}}[i,j] = \begin{cases} W_{\text{gate}}[i, k'a + r' + 1], & \text{if } j - 1 = k'b + r',\ k', r' \in \mathbb{N},\ r' < a,\ k'a + r' < d_{\text{ffn}}; \\ 0, & \text{otherwise}; \end{cases}$$

$$W'_{\text{up}}[i,j] = \begin{cases} W_{\text{up}}[i, k'a + r' + 1], & \text{if } j - 1 = k'b + r',\ k', r' \in \mathbb{N},\ r' < a,\ k'a + r' < d_{\text{ffn}}; \\ 0, & \text{otherwise}; \end{cases}$$

$$W'_{\text{down}}[i,j] = \begin{cases} W_{\text{gate}}[k'a + r' + 1, j], & \text{if } i - 1 = k'b + r',\ k', r' \in \mathbb{N},\ r' < a,\ k'a + r' < d_{\text{ffn}}; \\ 0, & \text{otherwise}; \end{cases}$$

Consider $f' = \text{MoC}_{a:b}(W'_{\text{gate}}, W'_{\text{up}}, W'_{\text{down}}) \in \mathcal{H}_{d_{\text{moc}}}^{a:b}$, we show that $f'(x) = f(x)$ for any $d$-dimensional row vector $x$. We define $g = xW_{\text{gate}}$, $u = xW_{\text{up}}$, $s = \text{SiLU}(g)$, $z = s \odot u$ such that $f(x) = zW_{\text{down}}$, and define $g' = xW'_{\text{gate}}$, $u' = xW'_{\text{up}}$, $s' = \text{SiLU}(g')$, $s'' = \text{TopK}_{a:b}(s')$, and $z' = s'' \odot u'$ such that $f'(x) = z'W'_{\text{down}}$. Here, $\text{TopK}_{a:b}$ preserves the $a$ entries with the largest absolute values in every consecutive block of $b$ entries and sets the rest to zero. For an input vector $x \in \mathbb{R}^{mb}$, this means that for $k' = 0, 1, \ldots, m-1$ and $r' = 1, 2, \ldots, b$:

$$\begin{aligned} &\text{TopK}a{:}b(x)[k'b + r'] \\ &= \begin{cases} x[k'b + r'], & \text{if } |x[k'b + r']| \text{ is among the top-}a \text{ in } |x[k'b+1]|, \ldots, |x[k'b+b]|, \\ 0, & \text{otherwise}. \end{cases} \end{aligned}$$

According to the definition of $W'_{\text{gate}}$, we have

$$
\begin{aligned}
g'[j] =& xW'_{\text{gate}}[:,j] \\
=& \begin{cases} xW_{\text{gate}}[:,k'a+r'+1], & \text{if } j-1 = k'b+r', \ k',r' \in \mathbb{N}, \ r' < a, \ k'a+r' < d_{\text{ffn}}; \\ 0, & \text{otherwise}; \end{cases} \\
=& \begin{cases} g[k'a+r'+1], & \text{if } j-1 = k'b+r', \ k',r' \in \mathbb{N}, \ r' < a, \ k'a+r' < d_{\text{ffn}}; \\ 0, & \text{otherwise}; \end{cases}
\end{aligned}
$$

Similarly, we have

$$
u'[j] = \begin{cases} u[k'a+r'+1], & \text{if } j-1 = k'b+r', \ k',r' \in \mathbb{N}, \ r' < a, \ k'a+r' < d_{\text{ffn}}; \\ 0, & \text{otherwise}; \end{cases}
$$

thus

$$
\begin{aligned}
s'[j] =& \text{SiLU}(g'[j]) \\
=& \begin{cases} \text{SiLU}(g[k'a+r'+1]), & \text{if } j-1 = k'b+r', \ k',r' \in \mathbb{N}, \ r' < a, \ k'a+r' < d_{\text{ffn}}; \\ 0, & \text{otherwise}; \end{cases} \\
=& \begin{cases} s[k'a+r'+1], & \text{if } j-1 = k'b+r', \ k',r' \in \mathbb{N}, \ r' < a, \ k'a+r' < d_{\text{ffn}}; \\ 0, & \text{otherwise}; \end{cases}
\end{aligned}
$$

Noting that for $\forall \, k' \in \{0,1,\cdots,k-1\}$, there are at least $(b-a)$ zero elements $s'[k'b+a+1]$, $s'[k'b+a+2], \cdots, s'[k'b+b]$ in the $b$ consecutive terms from $s'[k'b+1]$ to $s'[k'b+b]$, we have $s'' = s'$ since all non-zero elements of $s'$ is maintained in the $\text{TopK}_{a:b}$ selection. This indicates

$$
\begin{aligned}
z'[j] = s''[j] \cdot u'[j] =& s'[j] \cdot u'[j] \\
=& \begin{cases} s[k'a+r'+1] \cdot u[k'a+r'+1], & \text{if } j-1 = k'b+r', \ k',r' \in \mathbb{N}, \ r' < a, \ k'a+r' < d_{\text{ffn}}; \\ 0, & \text{otherwise}; \end{cases} \\
=& \begin{cases} z[k'a+r'+1], & \text{if } j-1 = k'b+r', \ k',r' \in \mathbb{N}, \ r' < a, \ k'a+r' < d_{\text{ffn}}; \\ 0, & \text{otherwise}; \end{cases}
\end{aligned}
$$

Consequently, we have

$$
\begin{aligned}
f'(x)[j] =& \sum_{i=1}^{d_{\text{moc}}} z'[i] \cdot W'_{\text{down}}[i,j] \\
=& \sum_{\substack{k',r' \in \mathbb{N}, r' < a \\ k'a+r' < d_{\text{ffn}}}} z[k'a+r'+1] \cdot W_{\text{down}}[k'a+r'+1,j] \\
=& \sum_{i=1}^{d_{\text{ffn}}} z[i] \cdot W_{\text{down}}[i,j] = f(x)[j],
\end{aligned}
$$

which implies $f(x) = f'(x)$. By the arbitrariness of $x$, we conclude that $f = f' \in \mathcal{H}_{d_{\text{moc}}}^{a:b}$, which completes the proof. $\qquad\square$

**Remark.** The Top-$K$ strategy used in the proof of Theorem 1 slightly differs from that described in Section 3.2. The proof selects entries with the largest absolute values of the SiLU outputs, whereas Section 3.2 adopts a more efficient approximation by selecting the top (non-absolute) values of the SiLU inputs. Since SiLU activations for negative inputs are close to zero, selecting top SiLU inputs serves as a good approximation to selecting the top absolute values of the outputs. We adopt this input-based selection in practice for improved computational efficiency.

## C    ADDITIONAL RESULTS AND EXPERIMENTAL DETAILS

### C.1    GENERALIZATION AND APPLICABILITY

**Downstream evaluations.** To thoroughly assess the generalization capabilities of MoC models, we use the lm-evaluation-harness framework (Gao et al., 2024) to evaluate their zero-shot performance

Table 6: Validation perplexity and memory consumption of pre-training GQA models on the C4 dataset.

| Model | Structure | Perplexity | Memory (GB) |
|---|---|---|---|
| LLaMA-130M | GQA | 24.02 | 53.9 |
| LLaMA-130M | GQA+MoC | 24.26 | **34.4** |
| LLaMA-350M | GQA | 18.51 | 52.9 |
| LLaMA-350M | GQA+MoC | 18.69 | **36.9** |

Table 7: Comparison of Mixtral with MoC-style FFN or vanilla FFN on the C4 dataset. We report validation perplexity alongside total memory usage, which includes model weights, gradients, optimizer states, and activations. Constant $K$ is the number of activated channels. We typically set $K = 0.5d_{\text{model}}$, which is 18.75% of of the channel dimension $d_{\text{ffn}} = 8d_{\text{model}}/3$.

| | 160M | 530M |
|---|---|---|
| Vanilla Mixtral | **23.77** (48.3G) | **18.65** (41.7G) |
| Mixtral+MoC | **24.44** (38.2G) | **18.88** (30.0G) |
| batch size per GPU | 256 | 128 |
| $K/d_{\text{model}}$ | 256 / 512 | 512 / 1024 |

across a range of NLP benchmarks. This standardized suite ensures reproducibility and enables consistent, reliable comparisons of model performance. Specifically, we select five representative tasks across four categories:

- **Natural language understanding:** MMLU (Hendrycks et al., 2021)
- **Reasoning:** ARC-Easy and ARC-Challenge (Clark et al., 2018)
- **Commonsense understanding:** PIQA (Bisk et al., 2020)
- **Truthfulness:** TruthfulQA (Lin et al., 2022)

Results are shown in Table 3 and discussed in Section 6.

**Applicability to additional base models.** To further assess the generalization of MoC across different model architectures and scales, we extend our study to LLaMA with Grouped Query Attention (GQA) (Ainslie et al., 2023), Qwen3 (Yang et al., 2025), and LLaMA-7B. As summarized in Tables 6 and 19, MoC demonstrates strong compatibility with these settings: it consistently preserves model performance while substantially reducing memory consumption. These results suggest that MoC is broadly applicable to modern LLM architectures beyond the standard LLaMA baseline.

**Compatibility with MoE architectures.** To further investigate activation sparsity in MoE models and demonstrate that MoC offers activation efficiency orthogonal to the MoE architecture, we pre-train the Mixtral model (Jiang et al., 2024)—a representative MoE-based large language model—on the C4 dataset and evaluate its perplexity on a held-out test set. Specifically, for the Mixtral+MoC variant, we replace each expert's MLP with a MoC-style feedforward network and compare its performance to the baseline after pre-training. We adopt an experimental setup consistent with that described in Appendix C.4, with full configurations detailed in Table 17. As shown in Table 7, Mixtral with MoC achieves comparable perplexity to the original model while significantly reducing memory consumption, confirming the viability of MoC within MoE architectures.

### C.2 COMPARISONS WITH RELATED APPROACHES

**Comparisons with ReLU and dReLU.** To further demonstrate the effectiveness of MoC, we extend our comparisons to several other sparsity-based baseline methods, including Top-$K$ gating with dReLU (Song et al., 2024) and standard ReLU activations. As shown in Table 8, both dReLU and ReLU lead to noticeable performance degradation, whereas MoC consistently achieves lower validation perplexity. This highlights the importance of MoC's design in preserving the representational capacity of the original architecture while introducing sparsity.

Table 8: Validation perplexity of LLaMA-based Transformer models pre-trained on the C4 dataset.

| Model Size | dReLU (Song et al., 2024) | ReLU | MoC |
|---|---|---|---|
| 130M | 26.34 | 28.98 | **24.02** |
| 350M | 20.09 | 19.07 | **18.57** |

Table 9: Validation perplexity of LLaMA-based Transformer models pre-trained on the C4 dataset.

| Model Size | BackRazor | MoC |
|---|---|---|
| 130M | 25.12 | **24.02** |
| 350M | 19.40 | **18.57** |
| 1B | 16.97 | **15.62** |

Table 10: Peak memory consumption and throughput comparison between MoC and pure GCP.

| Method | Peak Memory (GB) | Throughput (tokens/s) |
|---|---|---|
| FFN+GCP | 39.43 | 6160 |
| MoC+GCP | 41.90 | **7470** |

Table 11: Effect of top-$K$'s position on validation perplexity of different model sizes.

| Top-$K$ Position | MoC-60M | MoC-130M |
|---|---|---|
| Before SiLU | **30.59** | **24.02** |
| After SiLU | 30.92 | 25.45 |

Table 12: Effect of number of activated channels $K$ on MoC-130M's validation perplexity.

| # Activated Channels $K$ | Perplexity |
|---|---|
| 256 | 25.65 |
| 384 | 24.02 |
| 512 | 23.91 |

**Comparison with BackRazor.** We further compare MoC against BackRazor (Jiang et al., 2022), a memory-efficient pre-training algorithm that leverages a Top-$K$ sparsifier to compress activations stored for backpropagation. As shown in Table 9, MoC consistently outperforms BackRazor across different model scales (130M, 350M, and 1B parameters) on the C4 dataset. These results highlight that MoC not only provides substantial memory savings but also better preserves the model's representation ability during training.

**Comparison with gradient checkpointing.** To further highlight MoC's advantage in the memory–compute trade-off, we compare it against the pure gradient checkpointing (GCP) method. As discussed in Section 4, MoC already incorporates gradient checkpointing for selected activations; therefore, we denote it as *MoC+GCP* in this subsection. As shown in Table 10, when FFN+GCP and MoC+GCP exhibit comparable peak memory consumption, their throughput differs substantially: FFN+GCP achieves 6160 tokens/s, whereas MoC+GCP reaches 7470 tokens/s. This result demonstrates that MoC+GCP preserves memory efficiency while avoiding the severe throughput degradation inherent in pure recomputation-based approaches.

C.3    ABLATION AND ANALYSIS

**Effect of top-$K$ position.** In Table 11, we evaluate the performance of MoC-60M and MoC-130M by applying the top-$K$ operator either before or after the SiLU activation function. The results indicate that placing the top-$K$ operator before the activation generally yields better performance.

**Effect of the number of channels $K$.** We study how the number of activated channels $K$ affects MoC. A larger $K$ generally improves representational capacity and reduces perplexity, whereas a smaller $K$ yields higher sparsity, leading to greater memory savings and faster inference. Thus, choosing $K$ involves balancing performance and efficiency. To guide our setup, we conduct ablation

Table 13: Validation perplexity of MoC-350M pre-trained on the C4 dataset with different $K$ values.

| $K$ | Memory (GB) | Perplexity |
|-----|-------------|------------|
| 256 | 29.2 | 18.79 |
| 512 | 34.6 | 18.57 |
| 768 | 39.9 | 18.52 |
| 1024 | 52.5 | 18.26 |

Table 14: Inference latency ($\mu$s) of a single FFN layer across different batch sizes. "Relative Acc." stands for the relative acceleration ratio of MoC compared with FFN.

| Batch Size | Dense FFN | MoC | Relative Acc. |
|------------|-----------|-----|---------------|
| 1 | 316 | **191** | 1.65$\times$ |
| 2 | 360 | **227** | 1.59$\times$ |
| 3 | 360 | **227** | 1.59$\times$ |
| 4 | 357 | **230** | 1.55$\times$ |

Table 15: Training throughput and GPU memory usage for pre-training LLaMA-1B on a single NVIDIA A800 (80GB) GPU. "bsz" denotes batch size, and "OOM" stands for out-of-memory.

|  | LLaMA, bsz=64 | LLaMA, bsz=128 | MoC, bsz=64 | MoC, bsz=128 |
|--|---------------|----------------|-------------|--------------|
| Training Throughput (tokens/s) | 7639 | OOM | 7470 | **7714** |
| GPU Memory Usage (GB) | 56.6 | OOM | **41.9** | 73.7 |

experiments on different $K$ values with MoC-130M (Table 12) and MoC-350M (Table 13). The results confirm that performance improves as $K$ increases, but memory usage grows roughly linearly with $K$. Considering this trade-off, we adopt $K = 512$ for MoC-350M in our main experiments.

**Efficiency gains from reduced memory cost.** The memory reduction offered by MoC can translate into practical training efficiency gains. In particular, smaller memory footprints allow for larger batch sizes on the same hardware, which may improve throughput by reducing communication and synchronization overhead. As shown in Table 15, MoC enables pre-training LLaMA-1B with a batch size of 128 on a single NVIDIA A800 (80GB) GPU, a setting that vanilla LLaMA cannot support due to out-of-memory (OOM) errors. With this enlarged batch size, MoC achieves a training throughput of 7714 tokens/s, surpassing vanilla LLaMA's 7639 tokens/s. This demonstrates that MoC not only reduces memory consumption but can also unlock additional efficiency benefits during large-scale pre-training.

**Sparsity Patterns in MoE Models.** We observe that sparse activation is a widespread phenomenon in large language models, not limited to LLaMA-like architectures. To illustrate this, Figure 6 presents histograms of pre-SiLU and post-SiLU activations from various layers of LLaMA-MoE (Zhu et al., 2024b), a pre-trained MoE model. The activations in LLaMA-MoE exhibit significant sparsity, closely resembling the patterns observed in LLaMA, as discussed in Section 3.1.

**Batch size scaling.** We further investigate the inference-time acceleration of MoC under different batch sizes. As shown in Table 14, MoC consistently delivers substantial latency reductions compared to the dense baseline, even as the batch size increases from 1 to 4. These results demonstrate that the benefits of activation sparsity are not confined to the single-token decoding case, but also extend to small-batch scenarios that are common in practical auto-regressive inference.

**Implementation discussion.** From an implementation perspective, modern GPUs (e.g., NVIDIA architectures) comprise hundreds of independent Streaming Multiprocessors (SMs), each equipped with dedicated on-chip memory. During inference, different sequences within a batch can be dispatched to separate SMs, enabling each SM to load only the relevant (sparse) portions of the weight matrices. This strategy preserves sparsity, avoids redundant memory transfers, and underpins MoC's ability to sustain acceleration at small but non-trivial batch sizes.

## C.4 PRE-TRAINING DETAILS

**Experimental Setup.** For LLaMA pre-training across all model sizes, we follow the setup outlined in Zhao et al. (2024); Han et al. (2024). We use a total batch size of 512 and a maximum sequence length of 256, resulting in approximately 131K tokens per batch. The AdamW optimizer is employed with momentum parameters $(\beta_1, \beta_2) = (0.9, 0.999)$ across all experiments. The learning rate follows a cosine annealing schedule, preceded by linear warm-up during the first 10% of training steps.

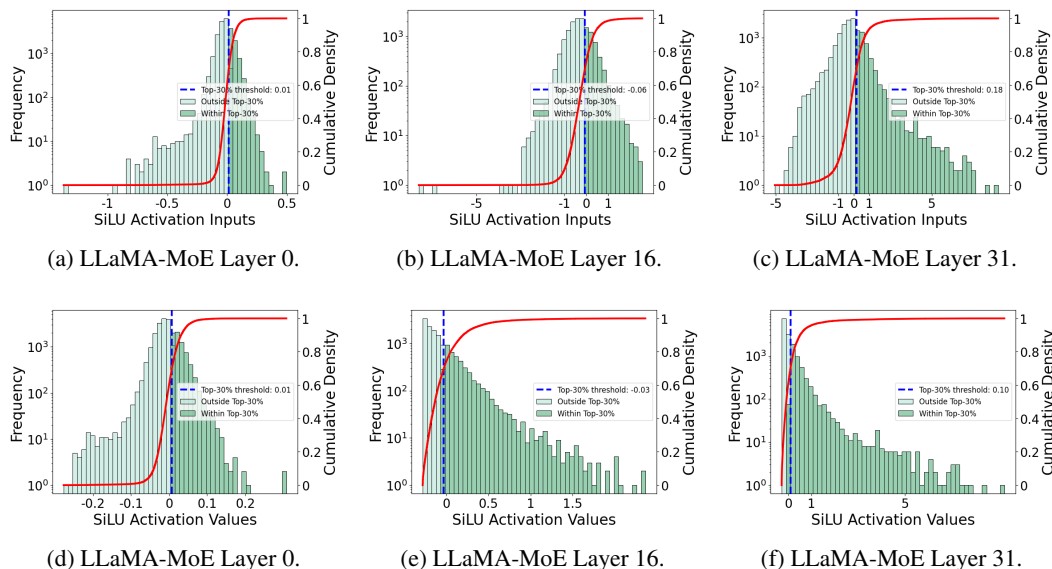

(a) LLaMA-MoE Layer 0.  (b) LLaMA-MoE Layer 16.  (c) LLaMA-MoE Layer 31.

(d) LLaMA-MoE Layer 0.  (e) LLaMA-MoE Layer 16.  (f) LLaMA-MoE Layer 31.

Figure 6: Histograms of pre-SiLU and post-SiLU activations from different layers of LLaMA-MoE (Zhu et al., 2024b). Subfigures (a), (b), and (c) correspond to the pre-SiLU activations, while subfigures (d), (e), and (f) show the post-SiLU activations. The blue dashed line marks the threshold for the top 30% of activations by value, and the red curve represents the cumulative distribution.

Table 16: Detailed configurations in each pre-training experiment.

| Params | Hidden | Intermediate | Heads | Layers | Training Tokens | Learning Rate |
|--------|--------|--------------|-------|--------|-----------------|---------------|
| 60M    | 512    | 1376         | 8     | 8      | 1.3B            | 2.5E-3        |
| 130M   | 768    | 2048         | 12    | 12     | 2.6B            | 2.5E-3        |
| 350M   | 1024   | 2736         | 16    | 24     | 7.8B            | 1E-3          |
| 1B     | 2048   | 5461         | 24    | 32     | 13.1B           | 6E-4          |

Table 17: Detailed configurations in pre-training experiments of Mixtral.

| Params | Experts | Hidden | Intermediate | Heads | Layers | Training Tokens | Learning Rate |
|--------|---------|--------|--------------|-------|--------|-----------------|---------------|
| 180M   | 2/8     | 512    | 1376         | 8     | 8      | 2.6B            | 2.5E-3        |
| 530M   | 2/8     | 768    | 2048         | 12    | 12     | 7.8B            | 1E-3          |

Table 18: Training throughput (samples/sec) on a single NVIDIA A800 GPU.

| Model + Optimizer | 350M | 1B |
|-------------------|------|-----|
| LLaMA (AdamW) | 28 830 | 5 975 |
| LLaMA (GaLore + AdamW) | 24 645 | 5 771 |
| MoC (AdamW) | 27 954 | 5 806 |
| batch size | 128 | 64 |

Table 19: Validation perplexity results of pre-training on the C4 dataset.

| Model | Perplexity |
|-------|------------|
| Qwen3-300M | 18.52 |
| Qwen3-300M+MoC | 18.59 |
| LLaMA-7B | 26.14 |
| LLaMA-7B+MoC | 26.47 |

Weight decay is set to 0. Detailed configurations and hyperparameters for each experiment are provided in Table 16.

**Training Efficiency.** We evaluate the training throughput of different methods by pre-training LLaMA-350M and LLaMA-1B with batch sizes of 128 and 64, respectively, on a single NVIDIA A800 GPU. Throughput is measured in tokens processed per second and averaged over 1,000 training

steps. As shown in Table 18, MoC achieves training efficiency comparable to the vanilla LLaMA model.

Notably, we hypothesize that $MoC_{2:8}$ can accelerate pre-training by leveraging the accelerated semi-sparse GEMM support available on recent NVIDIA GPUs, thanks to its sparse activation patterns. A thorough investigation of this potential remains beyond the scope of this study due to time constraints and is left for future work.

