# OpenReview forum: "Mixture-of-Channels: Exploiting Sparse FFNs for Efficient LLMs Pre-Training and Inference"
_ICLR.cc/2026/Conference — Submitted to ICLR 2026_

### Official Review · Reviewer_vhyn · 2025-10-26

**Soundness:** 3
**Presentation:** 3
**Contribution:** 2
**Rating:** 4
**Confidence:** 3

**Summary:**

Large Language Models (LLMs) have demonstrated remarkable success across various domains, driven by increases in both model and data scale. However, this growth also leads to significantly higher memory requirements. This paper focuses specifically on activation memory in feed-forward networks (FFNs). The authors propose Mixture of Channels (MoC), an FFN architecture that selectively activates only the Top-K most relevant channels per token, as determined by the native gating mechanism of SwiGLU. As a preliminary step to support their approach, the authors analyze the memory requirements of an LLM block and characterize the distribution of FFN activations. Additional sections detail the pre-training and inference procedures using MoC, followed by experiments evaluating MoC in terms of both accuracy and efficiency.

**Strengths:**

* The paper is clear and well structured.
* The empirical results present compelling evidence in support of the proposed method.
* Developing a kernel optimized for MoC on specific hardware is highly impressive. In many cases, the transition from a strong theoretical concept to a practical implementation ends at the stage of custom optimization. The fact that the authors went further to implement this kernel demonstrates a valuable contribution toward making MoC practically applicable.

**Weaknesses:**

* Theorem 1 isn't very clear - if $b\geq a$, $d_{moc}$ could exceed $f_{ffn}$ which would reduce efficiency. Claiming that MoC is as at least as good as dense FFN does not make a lot of sense. Typically, improving efficiency comes at the cost of reduced expressive power. An interesting direction would be to analyze how closely the model can approximate the original function given a certain efficiency constraint.

**Questions:**

* The primary motivation for MoC stems from the memory profiling presented in Section 2. However, this analysis was conducted on a relatively simple model, LLaMA 2. Would the observations differ if the same analysis were performed on a more recent model or one that incorporates a mixture-of-experts (MoE) module instead of a dense FFN?
* Inference efficiency - While it is true that for small batch sizes inference performance is primarily limited by I/O bounds, for larger batches it can become compute-bound and dominated by the FLOP count. Could the authors provide additional elaboration or analysis for this scenario?
* Can the authors provide additional downstream task evaluation to table 2 for a more complete image of the models comparison?
* The authors present their approach primarily as a pretraining method; however, an interesting alternative would be to apply MoC as a fine-tuning technique that adapts an already trained model to reduce its memory footprint. Using MoC in this way could also enable evaluation on larger-scale models (≥7B parameters).

---

### Official Review · Reviewer_8RoF · 2025-10-31

**Soundness:** 2
**Presentation:** 3
**Contribution:** 2
**Rating:** 4
**Confidence:** 4

**Summary:**

The paper proposed the mixture of channels framework, which preserves the top-k most active channels in FFN during training. Combined with the customized Triton kernel, it can save lots of GPU memory and speed up the end-to-end throughput by 1.13x. Across multiple scales, the MoC can achieve comparable ppl performance while significantly reducing the GPU memory.

**Strengths:**

1. The idea of inducing sparse computation during pretraining is reasonable, and the memory footprint is significantly reduced.
2. Experiments are conducted on diverse model structures.
3. The paper is well written and easy to follow.

**Weaknesses:**

1. The end-to-end latency speedup is fair on a single batch setting, which may also be minor under the batching inference.
2. The top-k-based training may induce instability due to the indifferentiable characteristics. I wonder how it compares with those softened masking methods.
3. There is no module-wise ablation study or profiling for the design kernel. I suggest a detailed profiling for the designed kernel.

**Questions:**

See weaknesses.

---

### Official Review · Reviewer_ZF1K · 2025-10-31

**Soundness:** 2
**Presentation:** 3
**Contribution:** 2
**Rating:** 4
**Confidence:** 3

**Summary:**

The paper proposes Mixture-of-Channels (MoC), a new feed-forward layer design that introduces activation sparsity within each Transformer FFN by selecting only the Top-K most activated hidden channels (neurons) for each token. The approach leverages the existing gating signal from SwiGLU to determine which channels are active, thereby reducing the number of intermediate activations stored and updated. The authors report reduced activation memory and improved training and inference throughput, maintaining decent model performance.

**Strengths:**

1. Clarity:
The paper clearly describes how MoC can be implemented as a more efficient FFN layer using existing SwiGLU gating values.
2. Efficiency benefit:
The approach can reduce activation and gradient memory during training and reduce inference latency on a single token.
3. Compatibility:
The proposed model can be integrated with modern optimized LLM kernels and systems.

**Weaknesses:**

1. Incremental novelty:
MoC is conceptually very close to CATS (Lee et al., 2024), differing mainly in using topK instead of threshold.
2. Fixed-K limitation:
The Top-K value is globally fixed across tokens, even though different tokens may require varying numbers of active channels. This may underutilize model capacity or cause redundancy for simple tokens.
3. Lack of accuracy validation:
The paper mainly evaluates on memory and throughput metrics. The datasets used are simple, and the models are small and outdated, and accuracy comparisons are missing for modern baselines (e.g., CATS (Lee et al., 2024), MoEfication (Zhang et el., 2021), Learn-to-be-Efficient (Zheng et al., 2024), etc.).

**Questions:**

How does MoC handle inference when different tokens in the same batch activate different channel subsets? Does this reduce inference efficiency?

---

### Meta-Review · Area_Chair_LcQs · 2025-12-29

**Summary:**

The paper proposes "Mixture-of-Channels" (MoC), a method designed to introduce sparsity into the Feed-Forward Networks (FFNs) of Large Language Models. By leveraging the inherent gating mechanism of SwiGLU, the method selectively activates only the Top-K most relevant channels for each token. The authors claim this approach significantly reduces activation memory footprints during pre-training and improves inference throughput by minimizing memory access requirements, supported by a custom Triton kernel implementation.

**Reviewer Concerns:**

The consensus among the reviewers is that while the problem of activation memory is relevant, the proposed solution lacks sufficient novelty and rigorous validation.

1. The most critical concern, primarily articulated by Reviewer ZF1K, addresses the incremental novelty and lack of comparative benchmarks. The reviewer pointed out that the MoC method is conceptually indistinguishable from the existing "CATS" framework (Lee et al., 2024), with the only differentiating factor being the use of a Top-K selection strategy instead of a threshold mechanism. Furthermore, the experimental section failed to compare MoC against relevant modern baselines such as CATS, MoEfication, or "Learn-to-be-Efficient," relying instead on outdated, small-scale models. Since the authors did not submit a rebuttal, this issue remains completely unresolved. From a reviewer's perspective, the failure to distinguish the work from prior art or empirically prove its superiority over the direct competitor (CATS) is a severe, likely fatal, flaw.

2. A second major area of concern is raised by all three reviewers. Reviewer 8RoF noted that the reported end-to-end speedup (1.13x) is marginal , while Reviewer vhyn correctly argued that for large batch sizes, inference becomes compute-bound (limited by FLOPs) rather than IO-bound, meaning the proposed memory reductions would not translate to significant latency gains. Additionally, Reviewer ZF1K questioned the engineering feasibility of handling batches where different tokens activate different channels, suggesting this could lead to inefficiency. The authors failed to provide the requested kernel profiling  or analysis for compute-bound scenarios. Consequently, the claim that this method is efficient for general "Inference" remains unconvincing to the reviewers, as the specific constraints (small batch vs. large batch) were not addressed.

3. Finally, Reviewer vhyn found the theoretical justification (Theorem 1) unclear and potentially flawed regarding efficiency bounds. Simultaneously, Reviewers ZF1K and 8RoF questioned the rigidity of a fixed global  value and the potential training instability arising from non-differentiable Top-K operations. The lack of a rebuttal means the authors missed the opportunity to clarify the theorem or provide ablation studies justifying the fixed-K design. Therefore, the reviewers' initial assessment that the method might be suboptimal or unstable stands firm.

**Reviewer Scores:**

Given the absence of an author response to address these substantial concerns, I anticipate the following score trajectory:

* **Reviewer ZF1K (Current: 4):** Likely to stay at **4** or drop to **2**. The identification of the CATS paper as prior art was a direct challenge to the paper's novelty. Without a counter-argument or new data, this reviewer will likely view the paper as redundant.
* **Reviewer 8RoF (Current: 4):** Likely to stay at **4** or drop to **2**. The reviewer was already skeptical about the marginal speedup and lack of profiling. The silence from the authors confirms the suspicion that the method does not scale well or offer significant benefits.
* **Reviewer vhyn (Current: 4):** Likely to stay at **4** or drop to **2**. The theoretical confusion and valid critique regarding compute-bound inference remain outstanding.

---

### Decision · Program_Chairs · 2026-01-26

Reject